# Towards Photonic Band Diagram Generation with Transformer-Latent Diffusion Models

## Abstract

Photonic crystals enable fine control over light propagation at the nanoscale, and thus play a central role in the development of photonic and quantum technologies. Photonic band diagrams (BDs) are a key tool to investigate light propagation into such inhomogeneous structured materials. However, computing BDs requires solving Maxwell's equations across many configurations, making it numerically expensive, especially when embedded in optimization loops for inverse design techniques, for example. To address this challenge, we introduce the first approach for BD generation based on diffusion models, with the capacity to later generalize and scale to arbitrary three-dimensional structures. This preliminary study couples a transformer encoder, which extracts contextual embeddings from the input structure, with a latent diffusion model to generate the corresponding BD. In addition, we provide insights into why transformers and diffusion models are well suited to capture the complex interference and scattering phenomena inherent to photonics. This cross-disciplinary approach is bridging modern deep learning architectures with complex photonic design problems, paving the way for new surrogate modeling strategies in this domain.

## 1 Introduction

Photonic crystals (PhCs) are periodic structures made of a repeated unit cell that can allow, alter or forbid light propagation. Designing new PhCs plays an important role in the development of technologies (Dhanabalan et al., 2023), notably in telecommunication (Russell, 2003; Katti & Prince, 2019) and optical computing (Yanik et al., 2003; Xu & Jin, 2023), with applications like low threshold lasers, low-loss resonators and cavities, efficient microwave antennas, and optical fibres (Soukoulis, 2001; Knight et al., 1999). For certain $(\omega, k)$ pairs, where $\omega$ is the frequency of light and $k$ its wavevector, light can become guided or even trapped within the designed PhC. This behavior is typically analyzed using band diagrams (BDs), like the ones illustrated in Figure 1. Indeed, BDs reveal the resonant modes, i.e., the specific frequency–wavevector combinations that are allowed within the inhomogeneous structured material. They are thus essential descriptors and play a key role in the development of novel applications with PhCs (Ozbay et al., 2004).

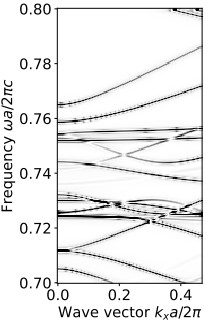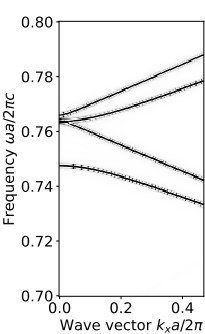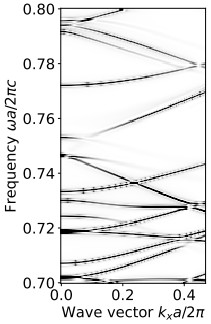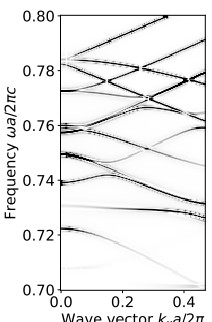

Figure 1: Examples of BDs obtained from rigorous coupled-wave analysis (RCWA) simulations. The $x$-axis represents the in-plane component $k_x$ of the wavevector of light, and the $y$-axis is the corresponding frequency $\omega$. A dark pixel indicates the existence of a mode for this pair of $(\omega, k_x)$.

While BDs can be seen as the identity card of PhCs, they usually require a significant amount of computation to be obtained. Indeed, each pair $(\omega, k)$ requires solving Maxwell's equations. This, in turns, leads to several thousands of simulations to compute a single BD. To fix ideas, computing a single BD with a $256 \times 128$ resolution roughly takes one minute using four CPU cores (4.7GHz), and can even take more than 10 minutes if using finer-grained settings[1]. When coupling this with an optimization process for inverse design, many problems can quickly become intractable.

Several recent works leverage the use of deep learning (DL) based surrogate models to solve this issue and support the work of physicists in photonics. However, none of them propose a method for photonic BD generation that is scalable to arbitrary three-dimensional (3D) structures. Furthermore, most of them focus on exploiting more standard architectures like dense networks, VAEs or U-Nets, without considering recent advancements in the DL community. This point is even more striking given that bypassing simulations in photonics is still considered a difficult task. While some analytical relations exist for one-dimensional PhCs, the complexity increases drastically in two dimensions, and becomes even more prohibitive in 3D (Yablonovitch, 1987; John, 1987; Ho et al., 1990; Joannopoulos et al., 2011). Wave propagation is affected by numerous constructive and destructive interferences caused by multiple scattering at layer interfaces and within local structures. Accurately capturing these potential high-order phenomena and their complex couplings across layers likely requires more advanced DL techniques.

In this work, we propose the first method to generate BDs directly from the input structures, which could later be used as a surrogate model to support photonics research. We combine (i) a latent diffusion model (Ho et al., 2020; Rombach et al., 2022) to generate the BD like images, with (ii) a transformer encoder (Vaswani et al., 2017) that generates meaningful conditioning embeddings from the geometric 3D structures to guide the diffusion process, and (iii) a VAE with a visual transformer encoder to map the BD to a latent space for the diffusion process. We found that transformer encoders naturally capture complex couplings via self-attention, and that their sequential formulation provides an elegant way to model 3D structures as sequences of 2D slices, avoiding the need for costly 3D convolutions. This slicing strategy makes our method easily extensible and scalable to arbitrary 3D photonic structures.

The contributions of this work can be summarized as follows:

(i) The first approach for BD generation based on diffusion models, with the potential to be extensible and scalable to arbitrary 3D photonic structures.

(ii) Demonstrating the use of transformer encoders in photonics to generate meaningful conditioning from the input structures, which could be transposed to other downstream tasks.

(iii) Providing insights on the reason why transformer encoders and diffusion models are good candidates for developing a new set of state-of-the-art surrogate models in photonics.

We believe that our work will shed light on the use of transformer encoders and diffusion models in the field of photonics, and will inspire other works in that direction. In this work, we therefore position our method explicitly as a proof-of-concept surrogate model for photonic research. Our main goal is to demonstrate that transformer-based conditioning and latent diffusion models can be used to generate meaningful photonic BDs. We also want to highlight that we do not aim to replace rigorous simulations such as RCWA or FDTD, but rather to provide a fast and approximate tool that can support and accelerate existing optimization pipelines.

In Section 2, we briefly describe some of the previous works using DL for photonics, to better position our work. Section 3 presents the photonic task that we are solving, as well as the dataset setup. Next, Section 4 details our strategy and the specific architectures we used, while giving motivations and insights about those choices. Section 5 presents the experimental setup and the results, which are after discussed in Section 6. Finally, Section 7 concludes this work.

## 2 RELATED WORK

The use of DL for photonics is currently an hot topic in this community, and has already been explored in several works (Peurifoy et al., 2018; Ma et al., 2021; Liu et al., 2021; Wiecha et al.,

---

[1]Considered settings are described later, and a more rigorous timing comparison is presented in Table 3.

2021; Chen et al., 2022). Among those works, Roy et al. (2024) use a U-Net surrogate model with particle swarm optimization for the design of vortex phase masks, bypassing the use of expensive Finite-Difference Time-Domain (FDTD) simulations, Colomina-Martínez et al. (2024) propose DL models for bending phenomena on holographic volume gratings, and Adibnia et al. (2024) show how DL could help for the inverse design of plasmonic ring resonator switches.

Another related line of work investigates operator-learning approaches for fast optical simulations. For example, Ma et al. (2025) and Zhu et al. (2024) propose neural operator architectures tailored to fast FDTD-like field prediction in complex photonic devices. These methods primarily target optical field distribution rather than BDs, but they share with our approach the idea to replace a large number of expensive numerical simulations with trained surrogate models. However, our work focus more on global descriptors than local field evolution.

Closer to the task that we are considering, Christensen et al. (2020) use CNNs and GANs for inverse design and band structure prediction with 2D PhCs. It is then followed by the work of Nikulin et al. (2022) that can be considered as its extension to the use of VAEs. More recently, Wang et al. (2024) developed a U-Net to predict dispersion relations for 2D PhCs. However, these works always exhibit at least one of the following drawbacks: (i) they only consider the use of basic DL techniques that are maybe not adapted to faithfully model complex photonic tasks, (ii) they only focus on 2D slices without straightforward ways to extend to 3D, or simply, (iii) they are not designed to directly provide a solution for a full BD generation.

Furthermore, previous works in other fields already implemented generative strategies using diffusion models. One can for example cite the work of Shmakov et al. (2023) in high energy physics, Shu et al. (2023) for fluid dynamics or also Alakhdar et al. (2024) for drug design. Regarding photonics, the only previous work investigating the use of diffusion model is the one of Sun et al. (2024). In their work, they use a diffusion model with contrastive language-image pre-training (CLIP) to map simple textual structure information to optical field distribution maps. However, this work focuses on field distributions while photonic BDs are more global descriptors. Moreover, the use of CLIP embeddings from simple textual descriptors limits seriously the possibility to extend this work to arbitrary 3D structures.

Thus, we introduce the first approach for BD generation of 3D PhCs based on diffusion models and transformer encoders. Attention is paid to keep the method as general as possible, enabling future generalizations and extensions.

## 3 TASK CONSIDERED AND DATASET GENERATION

**Task considered**  The task considered in this work is the generation of BDs of 3D PhCs structures. These crystals are usually defined by a unit cell with infinite periodicity in the *x*-*y* directions, while variations are allowed along the *z*-axis. By not enforcing full 3D periodicity, our approach is more general and closer to actual designs that can be fabricated (Joannopoulos et al., 2011).

To train our models, we first need to generate a dataset. We restricted structures to be a particular stacking of holey and uniform layers, as illustrated in Figure 2. Holey layers are defined by a single air hole inclusion with electric permittivity $\epsilon = 1$ in the unit cell which may either be cell-centered, border-centered, or corner-aligned to allow different types of stacking configurations. The dielectric in each layer is restricted to $\epsilon$ values in $[4.0, 6.0]$. The hole radius is in $[0.1, 0.45]$ and the layer thickness can be in $[0.1, 0.3]^2$. Those values have been chosen for the proof-of-concept and because they were considered as relevant for experts in photonics. Also, the restriction to those particular intervals should not be seen as assumptions for our method to work.

**Dataset generation**  Each holey layer is always followed by a uniform layer with the same material and thickness. Material dielectric, hole radius, thickness and hole alignment are randomly sampled from the previous intervals. Based on those settings, we derived two different datasets. The first one (later referred to as the small one) is the easiest one and is made of 30,000 PhCs with one or two of the holey-uniform stack (leading to crystals with either 2 or 4 layers). The second one (later referred

---

[2] Note that all dimensions are given in reduced units, because we assume a unit cell of dimensions $a \times a$ in the $xy$ plane. For convenience, and because light-matter interactions of interest only occur when observed at the same scale, frequencies are also expressed in reduced units ($\tilde{\omega} = \frac{\omega a}{2\pi c}$).

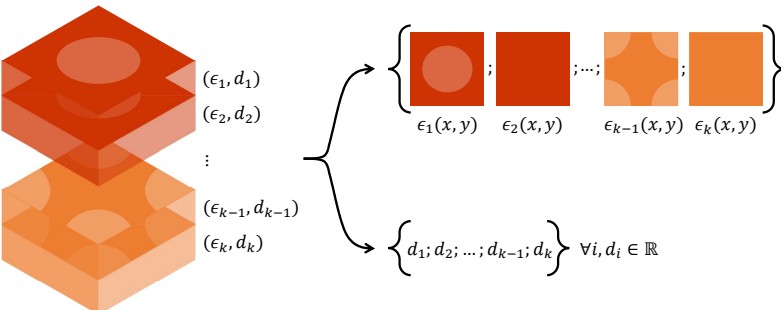

Figure 2: 3D structures are represented as ordered pair sequences $\{(\epsilon_1, d_1), \ldots, (\epsilon_k, d_k)\}$, where $\epsilon_i(x, y)$ are 2D dielectric maps for each layer and $d_i$ their corresponding thicknesses.

to as the large one) is made of 60,000 PhCs that can go up to 4 stacks (leading to crystals made of up to 8 layers). BDs are computed using rigorous coupled-wave analysis (RCWA) (Moharam & Gaylord, 1981; Li & Lin, 2003), for 256 frequencies between 0.7 and 0.8 (0.75 for the second dataset), and 128 values for $k_x$, leading to 32,768 RCWA simulations for a single BD. Each layer is represented on a $256 \times 256$ grid for the RCWA simulations, but resized and saved to only $64 \times 64$ pixels for the dataset. Indeed, while it is important to maintain enough spatial resolution for RCWA simulations, $64 \times 64$ pixels is enough to faithfully render the layers for a surrogate model. To ease the training process, BDs are also binarized and smoothed using an adaptive thresholding.

Despite the drastic choices made to restrict the design and simulations, we want to highlight that photonic BDs generation remains highly challenging. Even with apparently "simple" stacks of holey and uniform layers, the resulting BDs can exhibit strong variations and discontinuities along $\omega$. Small modifications in layer thickness, hole radius, or alignment may drastically impact the global BD, reflecting the highly non-linear and non-local nature of the mapping. It also underlines the difficulty of learning a reliable surrogate model. Moreover, a photonic BD is by definition fine-detailed, i.e., composed of fast and sometimes sparse variations along the $\omega$ axis. These points altogether are the reasons why more standard architectures may struggle to solve this task. At the same time, the restrictions should not be seen as a limitation of our approach. The method that we introduce in Section 4 is general and only requires to represent 3D PhCs as sequences of 2D slices. Nothing prevents its direct application to more complex geometries (e.g., arbitrary inclusions). The present dataset serves primarily as a controlled and computationally feasible benchmark to validate our framework as a proof-of-concept.

## 4 METHOD

In this section, the method used to generate BDs from the 3D PhCs is presented. The method can be decomposed into three main components that are described in this section: a material-to-context encoder, a BD encoder/decoder and a latent diffusion model.

It is worth mentioning that prior to the following method, we explored a GAN-based pipeline. While this approach showed some potential, it did not yield convincing results. We also experimented with 3D architectures such as 3D-CNNs/3D-VAEs that directly operate on full 3D dielectric tensors. However, these models quickly became memory-intensive for realistic resolutions and depths and, in our experiments, either failed to converge or produced blurred BDs. By contrast, representing the structure as a sequence of 2D slices and processing it with a transformer encoder provided a much more stable and scalable way to capture long-range inter-layer couplings. This motivated our use of diffusion models and transformer encoders. Diffusion models are known to be able to generate fine-detailed outputs (Rombach et al., 2022). When combined with transformers and suitable contrastive training, those models showed themselves to be much more adapted to the task. From those observations, we decided to focus this paper on the presentation of our working and original pipeline, considering it as a preliminary work. However, for completeness, we added some details in Appendix C about the GAN-based approach that we tested on the exact same setup, as well as the results we obtained with.

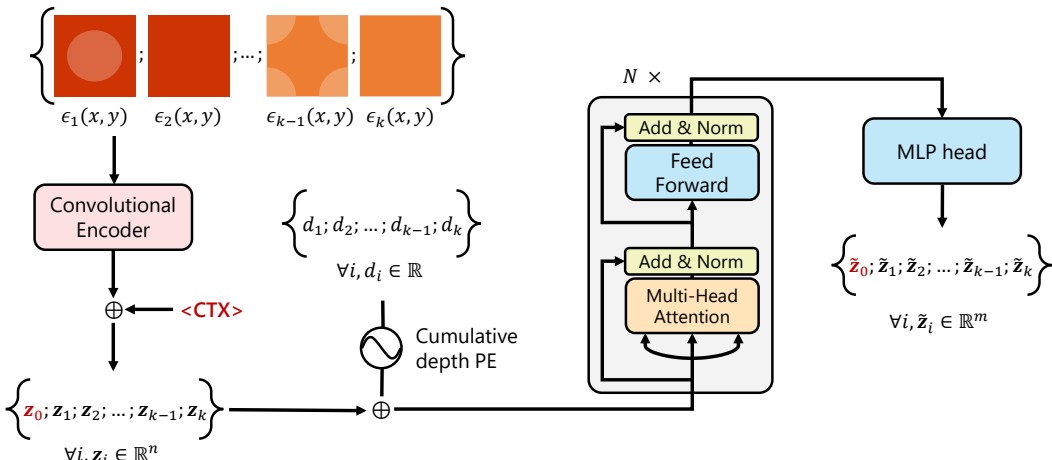

Figure 3: M2C encoder architecture — A sequence of $k$ dielectric layers $\{\epsilon_1(x, y), \ldots, \epsilon_k(x, y)\}$ is encoded by a shared convolutional encoder, and concatenated with a context token. This yields a sequence of latent vectors $\{\mathbf{z}_0, \mathbf{z}_1, \ldots, \mathbf{z}_k\}$, each in $\mathbb{R}^n$. Cumulative depth positional encodings (PE) are added to preserve both the ordering and the thicknesses of each layer. The sequence is then processed by $N$ stacked transformer encoder blocks. Finally, a MLP head projects the hidden dimension $n$ to the output dimension $m$, producing the contextual representations $\{\tilde{\mathbf{z}}_0, \tilde{\mathbf{z}}_1, \ldots, \tilde{\mathbf{z}}_k\}$, where $\tilde{\mathbf{z}}_0$ serves as the global contextual embedding.

### 4.1 MATERIAL-TO-CONTEXT (M2C) ENCODER

At the core of our method lies the idea that transformer encoders are particularly well suited to deal with photonics. In photonics, the global optical response of a material often results from complex couplings between distant layers, due to multiple reflections such as Fabry–Pérot or Fano effects (Saleh & Teich, 1991; Limonov et al., 2017), or higher-order scattering phenomena. Self-attention naturally models such interactions by allowing each layer representation to attend to all others, while multi-head attention can disentangle distinct coupling mechanisms. These abilities motivate our use of a transformer encoder as the backbone of the material-to-context (M2C) component.

The goal of the M2C encoder is to replace the classical CLIP model for textual conditioning. The M2C encoder will map an input structure into a contextual embedding that can later guide the diffusion model. Its architecture is presented in Figure 3. As previously explained, each 3D structure is represented by a stacking of 2D dielectric slices and their corresponding thicknesses. Each slice is first concatenated with the squared modulus of its 2D Fourier transform along the channel dimension to provide spectral information and inform of periodicity. After that, a shared convolutional encoder is used to extract structural features from the slices. The resulting embeddings are then flattened into a vector representation. To preserve the ordering of layers and their physical thicknesses, we introduce a cumulative depth positional encoding that is added to each layer embedding. This ensures that the model distinguishes not only the content of each slice but also its position and thickness. To the sequence of layer embeddings, we prepend a learnable context token (CTX), which is designed to aggregate global information about the full structure and later devoted to contrastive alignment. The extended sequence is processed by a stack of transformer encoder blocks, following the standard encoding path of transformer networks, as proposed by Vaswani et al. (2017). A final MLP head is then used to project the embeddings from the transformer encoder to the desired final dimension. The context token is considered as a compact, global representation of the entire structure, while the other outputs are devoted to retain more slice-specific information.

### 4.2 BAND DIAGRAM (BD) ENCODER/DECODER

The objective of the BD encoder/decoder is twofold: firstly, it enables the use of a latent diffusion process by mapping BDs into and reconstructing them from a lower-dimensional latent space. Sec-

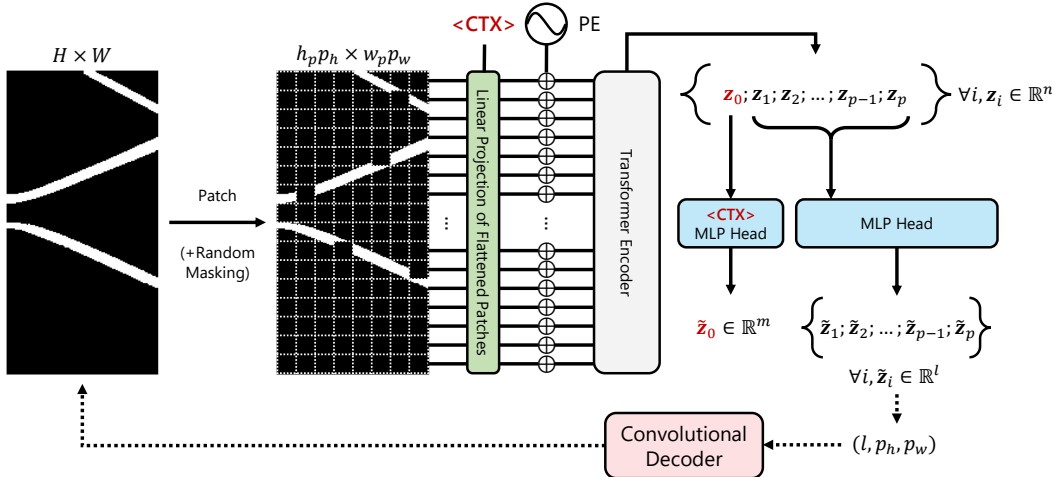

Figure 4: BD encoder/decoder architecture — A BD of size $H \times W$ is divided into $p = p_h p_w$ patches (optionally with random masking during training), each flattened and linearly projected. A context token is concatenated to the sequence, and positional encodings (PE) are added to preserve spatial relationships between patches. The resulting sequence $\{\mathbf{z}_0, \mathbf{z}_1, \ldots, \mathbf{z}_p\}$ with $\mathbf{z}_i \in \mathbb{R}^n$, is processed by $N$ stacked transformer encoder blocks (similarly to the M2C encoder architecture). The context token is then mapped by an MLP head to a global representation $\tilde{\mathbf{z}}_0 \in \mathbb{R}^m$, while the patch embeddings are projected to local representations $\{\tilde{\mathbf{z}}_1, \ldots, \tilde{\mathbf{z}}_k\}$ with $\tilde{\mathbf{z}}_i \in \mathbb{R}^l$. These local embeddings can then be reshaped into a single latent representation with shape $(l, p_h, p_w)$, to later be forwarded to a convolutional decoder trained to reconstruct the BD in pixel space.

ondly, it provides a global representation of each BD, which is used for contrastive training with the M2C encoder. The global architecture of the BD encoder/decoder is presented in Figure 4.

For this model, we design a variational autoencoder (VAE) in which the encoder path is implemented as a vision transformer (ViT). A BD image is divided into non-overlapping patches, each of which is linearly projected into a patch embedding. A learnable context token is also prepended to the sequence, and positional encodings are added to preserve the spatial arrangement of the patches. The sequence is then processed by a stack of transformer encoder blocks. This architecture provides two key benefits. Firstly, the global embedding obtained from the context token is used as a compact BD representation for contrastive alignment. Secondly, the patch embeddings are reshaped into a structured latent tensor that is decoded back into pixel space by a convolutional decoder, following the standard decoding path of a VAE. Using a ViT instead of a CNN for the encoding path is particularly well motivated in this context: each patch corresponds to a localized region in the frequency–wavevector domain, making it natural to treat BDs as sequences of spectral compartments. While convolutions tend to mix information across neighboring regions, a transformer will first consider them independently and learn correlations between spectral windows. An additional advantage of the ViT-based encoder is the possibility to mask some patches during training. By randomly dropping a fraction of input patches, the model is encouraged to reconstruct missing regions of the BD, leading to improved robustness and generalization.

### 4.3 BAND DIAGRAM (BD) DIFFUSION PROCESS

The final stage of our pipeline is a latent diffusion model that generates BD images conditioned on the material embedding, following the image generation strategy from Rombach et al. (2022), and presented in Figure 5. During training, the BD encoder first maps the target BD into a latent representation, which is iteratively perturbed with Gaussian noise following the forward diffusion process. At each diffusion step $t$, a denoising U-Net is trained to predict the added noise, conditioned on the global structural embedding from the M2C encoder. Once trained, the model can sample BDs by starting from pure noise and iteratively reversing the diffusion process. The recovered latent representation is then decoded by the BD decoder to reconstruct the BD in pixel space.

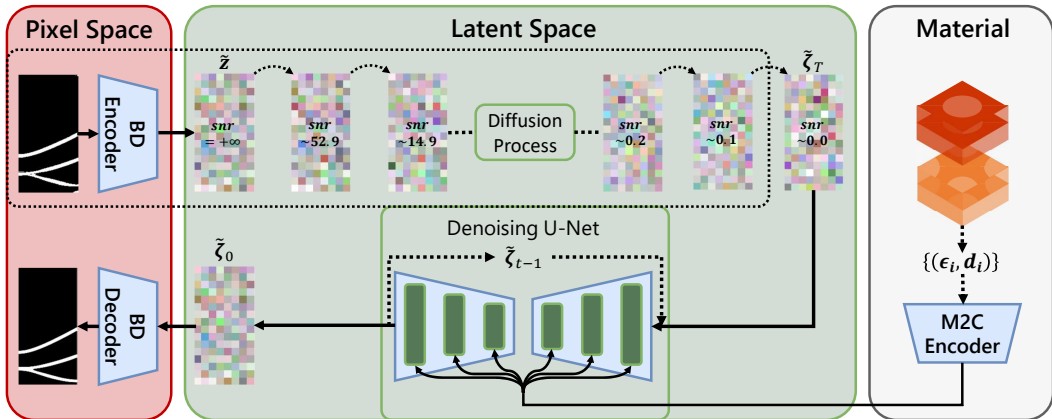

Figure 5: The local latent representations are first extracted from the target BDs thanks to the BD encoder, and reshaped to a latent representation $\tilde{\mathbf{z}}$ of shape $(l, p_h, p_w)$. A forward diffusion process iteratively perturbs these latents with Gaussian noise, progressively reducing the signal-to-noise ratio ($snr$). A denoising U-Net, conditioned via cross-attention on the material representation from the M2C encoder, is trained to iteratively predict the noise at each time step $t$. After reversing the diffusion process, the recovered latent $\tilde{\zeta}_0$ is forwarded to the BD decoder to reconstruct the BD. The steps highlighted with the black dotted box are only performed for training. During inference, the process directly starts from pure random noise. Figure inspired from Rombach et al. (2022).

## 5 EXPERIMENTS AND RESULTS

This section is divided in two main parts. The first focuses on the training of the M2C and BD encoders/decoder, as well as the evaluation of their standalone performances. The M2C encoder, in particular, is designed to capture as much information from the structure as possible and could also support other downstream tasks beyond BD generation in the future. The second part focuses on BD generation using the latent diffusion model, providing both qualitative and quantitative results.

**The M2C and BD encoders/decoder** are jointly trained using a contrastive method for the alignment of the context tokens following the idea of momentum contrast (MoCo) (He et al., 2020). This first training stage allows aligning the structural information with its corresponding BD by enforcing a shared latent space between the two models. To do so, for each pair of input structure and BD in a batch, the M2C and BD encoders are optimized such that the similarity (dot product) between the global contextual embeddings is maximal when the structure is compared to its corresponding BD, and minimal otherwise. Still following the initial paper of MoCo, a strategy of online and offline weights is setup with exponential moving average (EMA) to feed a First-In First-Out memory bank to artificially increase the number of negative pairs per batch. Simultaneously, the BD decoder is trained to minimize a binary cross entropy reconstruction loss combined with KL regularization on the latent distribution, ensuring that the latent space preserves sufficient information to reconstruct the BD. This dual training strategy provides strong alignment between the M2C encoder and the BD encoder such that the M2C contextual representation becomes a meaningful conditioning for the diffusion model. The final output dimension for the conditioning embeddings (MLP heads) is fixed to 768, to match the usual dimension of conditional vectors in diffusion models. Cumulative depth positional encodings are implemented using a trainable MLP, as this solution led to better results.

During training, a global and patch dropout of $0.2$ are used along with two data augmentation strategies. The first one randomly adds one or several layers of air ($\epsilon = 1$) at the beginning and/or the end of a structure. Indeed, adding such layers should not have any impact on the final BD. The second one performs a random roll for all the layers across the structure. Again, because of the periodicity of the unit cells in the $xy$ plan, this modification should not have any impact on the final BD.

To evaluate the performance of the M2C and BD encoders/decoder, the architectures are trained on both the small and large datasets while keeping $20\%$ of the data for testing. For each one of them, the top-1 and top-5 testing accuracy are reported. Concretely, for each batch, for each structure, all BDs are ranked by similarity. The top-1 accuracy is the fraction of structures for which the true

Table 1: Architectures for the M2C and BD encoders/decoder and their contrastive performances. $N$ is the number of transformer blocks, $d_{model}$ their dimensionality, $d_{ff}$ the size of the dense layers and $h$ the number of attention heads. Top-1 and 5 accuracy are reported for the testing sets of both datasets, resp. with the smaller (i) and larger (ii) stacks.

| M2C encoder | | | | BD encoder / decoder | | | | Small dataset | | Large dataset | |
|---|---|---|---|---|---|---|---|---|---|---|---|
| $N$ | $d_{model}$ | $d_{ff}$ | $h$ | $N$ | $d_{model}$ | $d_{ff}$ | $h$ | Top-1 (%) | Top-5 (%) | Top-1 (%) | Top-5 (%) |
| 16 | 256 | 1024 | 16 | 12 | 128 | 1024 | 16 | 79.79 | 96.70 | 57.63 | 84.46 |

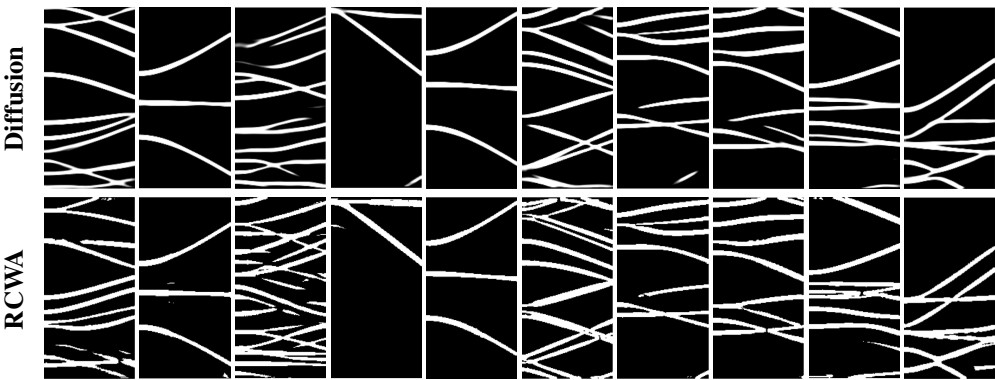

Figure 6: Qualitative comparison of BD generation for the small dataset. The first row shows BDs generated with the diffusion model, and the second one, their corresponding ground truth from RCWA simulations. Samples are randomly selected from the testing set.

BD ranks first, while the top-5 counts the matching correct if it is at least in the five most similar candidates. Note that those metrics are computed in both directions (BD → M2C and M2C → BD) and averaged. The configuration and associated results are presented in Table 1. A description of the training settings, as well as more details about the architectures is presented in Appendix A.

**The latent diffusion model** is a standard U-Net with residual and attention blocks. Conditioning comes from two sources: the M2C encoder output, injected through cross-attention, and the denoising time step, provided as 1280-dimensional embeddings and added within the residual blocks.

The diffusion process is defined with a cosine noise scheduler over $1,000$ steps. Training relies on a signal-to-noise ratio weighted MSE loss between the true and predicted noise. To reflect the one-to-one mapping between structures and BDs, we also considered a binary cross-entropy loss between the generated and ground-truth BDs. In addition, we experimented with freezing or fine-tuning the M2C encoder, allowing it to adapt if needed. For sampling, we use the denoising diffusion implicit models (DDIM) method introduced by Song et al. (2021) that reduces the number of required steps to generate an image. Instead of reversing the full $1,000$ diffusion steps, we generate BDs in 50 steps. More details about the architecture and the training parameters can be found in Appendix A.

Figure 6 shows qualitative examples from our best-performing model for the small dataset. More qualitative examples, including those for the large dataset, can be found in the Appendix B. Quantitative results, including dice coefficient and mean average precision (mAP), are reported in Table 2. Finally, Table 3 compares computation times across different methods.

## 6 DISCUSSION AND LIMITATIONS

Firstly, regarding the M2C and BD encoders/decoder, we observe very strong overall performance. With the batch size of 128, random ranking would yield only $0.78\%/3.90\%$ top-1/5 accuracy. In contrast, our models achieve $79.79\%/96.70\%$ on the small dataset and $57.63\%/84.46\%$ on the large one, clearly demonstrating the capacity of the encoders to extract meaningful features.

Table 2: Quantitative results with different training strategies. The effect of fine-tuning the M2C encoder and adding a reconstruction loss is evaluated. Performances are reported with dice coefficient and mean average precision (mAP) over the testing set, for both datasets.

| Finetuned M2C | Reconstruction loss | Small dataset | | Large dataset | |
|:---:|:---:|:---:|:---:|:---:|:---:|
| | | Dice | mAP | Dice | mAP |
| ✗ | ✗ | **0.374** | **0.437** | 0.228 | **0.238** |
| ✓ | ✗ | 0.231 | 0.230 | 0.208 | 0.199 |
| ✗ | ✓ | 0.370 | 0.435 | **0.232** | 0.236 |
| ✓ | ✓ | 0.229 | 0.229 | 0.208 | 0.199 |

Table 3: Comparison between computation times (in seconds) for the diffusion model and RCWA simulations. For each fixed number of layers, mean and standard deviation values are reported through 10 different structures. RCWA simulations are performed using 4 CPU cores (4.7GHz) while considering $3 \times 3$, $5 \times 5$, and $7 \times 7$ plane waves, covering various simulation precision. Diffusion inference is tested both on CPU and GPU (NVIDIA RTX 3070 8GB).

| # Layers | Diffusion | | RCWA | | |
|:---|:---:|:---:|:---:|:---:|:---:|
| | CPU | GPU | 3×3 | 5×5 | 7×7 |
| 4 | $2.65 \pm 0.06$ | $1.19 \pm 0.02$ | $67.46 \pm 2.63$ | $161.69 \pm 5.06$ | $617.40 \pm 6.01$ |
| 8 | $2.64 \pm 0.07$ | $1.18 \pm 0.02$ | $97.07 \pm 12.66$ | $260.74 \pm 7.91$ | $1063.14 \pm 5.67$ |

Secondly, results for the photonic BD generation task are highly encouraging. Qualitatively, the generated structures are generally in good agreement with the simulations, as it can be observed in Figure 6. Quantitatively, from Table 2, we find that finetuning the M2C encoder during training actually degrade the performance compared to fully freezing it after the contrastive pretraining. In addition, the reconstruction loss previously motivated does not seem to provide complementary information to the standard noise prediction loss. Regarding the quantitative results, one should also add that because photonic band diagrams consist of very thin and sometimes sparse pixel bands, these overlap-based metrics are difficult to interpret in absolute terms compared to more traditional computer vision tasks. Indeed, even a small spatial shift of a few pixels can cause a large drop in Dice or mAP. In this work, we therefore use these scores primarily as relative indicators to compare different training strategies and architectures. For assessing the final performance, qualitative inspection of the generated BDs remains more informative than the raw metric values.

Nevertheless, BD generation becomes more challenging as the number of modes (bands) increases. This may be physically explained by the fact that wave propagation inside the corresponding structures is more complex. Consequently, results on the large dataset, where more bands are typically visible, are less accurate both qualitatively and quantitatively. We hypothesize that further improving the M2C encoder could help mitigate this limitation. In preliminary experiments, we consistently observed that higher contrastive alignment accuracy between the M2C and BD encoders translates into more faithful BD generation. In other words, improvements in the contrastive pretraining stage directly correlate with qualitative and quantitative gains for the diffusion model. Possible directions include increasing the size of the network to capture more complex layer couplings, or incorporating physics-informed strategies like proposed by Chen et al. (2020); Lu et al. (2021); Cuomo et al. (2022); Lim & Psaltis (2022); Tsakyridis et al. (2024); Medvedev et al. (2025). Another idea worth exploring is to draw multiple diffusion samples from different random noise initializations for the same structure. Aggregating these samples would not only allow us to refine the final BD (e.g., by averaging or denoising inconsistencies), but also to build pixel-wise variance or confidence maps that quantify the model uncertainty over the spectral domain.

Despite the fact that BD generation does not yet reach the physical fidelity of real simulations, Table 3 highlights how fast it can be. Compared to basic $3 \times 3$ RCWA simulations, our surrogate models achieve speedups of $56\times$ to $82\times$. In addition, inference time does almost not depend to the number of layers, unlike physical simulations. Diffusion models thus provide a practical tool for fast exploration of large design space. Finally, it is worth noting that the simulations used to build our datasets were performed with a lightweight setting for proof-of-concept (RCWA with $3 \times 3$

plane waves), meaning that those previous speedups are largely underestimated. As also presented in Table 3, finer-grained simulations (e.g., $7 \times 7$ plane waves) can be one order of magnitude slower.

## 7 CONCLUSION

We introduce a new method that leverages latent diffusion models and transformer encoders to generate band diagrams (BDs) for 3D PhCs. While our dataset covers only a limited set of structures, the method is designed to be generalizable and scalable to more arbitrary 3D configurations.

The transformer encoders prove to be a powerful and expressive approach for tackling photonic problems. After contrastive training, we achieved a top-5 accuracy of approximately $97\%$ in similarity ranking, indicating that the M2C encoder successfully learned a meaningful embedding for the structures. This strong performance may be explained by the self-attention mechanism, where each layer iteratively attends to all others, making the modeling of scattering, interference, and other physical phenomena more explicit. As a perspective, the idea behind the M2C encoder and the whole pipeline could be transposed to other tasks in physics, even beyond the particular context of photonics (like the similar problem of electronic band structures in solid-state physics, for example).

Also, we successfully trained latent diffusion models to generate BDs, achieving promising results that highlight their potential as surrogate models for inverse design. Notably, the models provide a speedup of up to $56\times$ for small structures and up to $82\times$ for the larger ones when compared with RCWA simulations using only $3 \times 3$ plane waves. When considering more detailed $7 \times 7$ simulations, the speedup factors can reach $518\times$ and $900\times$, respectively.

Despite these encouraging results, some limitations currently remain and pave the way to further works in this direction. In particular, the fidelity of the generated BDs compared to real simulation is inevitably reduced, especially when increasing the number of layers. Nevertheless, this limitation should also be balanced by the fact that surrogate models should not fully replace rigorous simulations. Its main value lies in providing fast, and approximate feedback during exploration process, while high precision simulations remain essential for final validation. In this role, the substantial speedups make them highly valuable for fast exploration during inverse design, for example.

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

## A  TRAINING AND ARCHITECTURAL DETAILS

This appendix presents the training settings used for the M2C and BD encoders/decoder contrastive alignment, as well as the training of the latent diffusion models. In addition, more architectural details that are not directly presented in the core of the paper are presented, for reproducibility.

All the trainings are performed on NVIDIA A100 (40GB), with PyTorch 2.7.1 and CUDA 12.6.

### A.1  M2C AND BD ENCODERS/DECODER CONTRASTIVE ALIGNMENT

For the joint training of those models, the batch size is fixed to 128. The AdamW optimizer is used with a learning rate of $10^{-4}$ and a weight decay of $2 \times 10^{-4}$ through 500 epochs. Note that this weight decay is not applied for the trainable positional encoding, as it can lead to instabilities during training. A cosine annealing learning rate scheduler is used with a linear warm-up of 50 epochs. The KL regularization is weighted by a factor $10^{-3}$ when added to the contrastive and reconstruction losses. The contrastive loss is weighted by a coefficient $\alpha$ that follows a linear warm-up schedule:

$$\alpha = \begin{cases} \dfrac{e+1}{E_{\text{warmup}}}, & \text{if } e < E_{\text{warmup}}, \\ 1.0, & \text{otherwise}, \end{cases}$$

where $e$ denotes the epoch and $E_{\text{warmup}}$ is the number of warm-up epochs. The reconstruction term uses a weight of 1.0, resulting in a full training loss

$$\mathcal{L} = \alpha \mathcal{L}_{\text{contrastive}} + \mathcal{L}_{\text{recon}} + 10^{-3} \mathcal{L}_{\text{KL}}.$$

Regarding the contrastive setup, we set the size of the memory bank to 131,072, and the exponential moving average (EMA) weight is set to 0.999. A standard temperature parameters is defined and initialized with a logit scale of 0.07, which corresponds to typical temperatures used in MoCo-style contrastive learning. This scale is left trainable and clipped to the interval $[0, 3.4]$, which matches common values observed in contrastive setups.

### A.2  MORE DETAILS ABOUT THE M2C ENCODER

The M2C encoder is made of four main components: a convolutional encoder, a cumulative depth positional encoding, a transformer encoder, and a final MLP head.

The convolutional encoder is a standard CNN, following the standard architecture of the encoder path of a VAE. It is made of residual (each made of two convolutional layers) and attention blocks, similar to the ones used in the U-Net for the latent diffusion model. A first convolution is used to merge the spatial and spectral features (from the 2D-FFT). After that, three downsampling blocks made of two residual blocks and a downsampling convolution with stride $2 \times 2$ are defined, with 32, 64 and 128 output filters, respectively. A bottleneck is defined afterwards with several four residual blocks and an attention blocks sandwiched in between. Attention pooling is performed at the end to convert the input images to one-dimensional embeddings.

The cumulative depth encoding is defined by a simple MLP made of two dense layers with 512 and 256 output neurons, respectively, with GELU activation function and layer normalization in between. This layer takes the sequence of cumulative thicknesses and output a sequence of 256-dimensional embeddings, devoted to help the transformer encoder to better discriminate the position and thickness of each layer.

Finally, the M2C encoder ends with a standard transformer encoder defined by the parameters presented in Table 1, and a final MLP head, made of one simple dense layer, projecting the embeddings into a 768-dimensional space.

All of these leads to a M2C encoder with $\sim 17.3$ millions parameters.

### A.3 More details about the BD encoder/decoder

The BD encoder starts with a ViT. This ViT splits the input images into patches of dimensions $16 \times 16$. Those $\frac{256}{16} \times \frac{128}{16} = 16 \times 8 = 128$ patches are then flattened and forwarded to a dense layer (with layer normalization) to be projected to a sequence of 128-dimensional embeddings. The parameters for the transformer encoder that follows are presented in Table 1.

After the ViT, the sequence is split into two directions, with two different purposes. The first one takes the context token (prepended previously, before the linear projections), and forward it to a MLP head to project the input to a final 768-dimensional embedding (for contrastive alignment with the M2C encoder). The second one first project the embeddings into two distinct 4-dimensional embeddings to later mimic a standard VAE (resp., the mean and standard deviation tensors). The resulting $128 \times 4$ embeddings are reshaped into latent images of shape $4 \times 16 \times 8$.

Finally, the latent images are forwarded to a convolutional decoder, made again of residual and attention blocks, followed by several upsampling operations. Even though the operations are linked together, the convolutional decoder mainly follows a symmetric path to the M2C convolutional encoder (replacing the down convolutions by upsampling convolutions).

The final BD encoder/decoder is made of $\sim 7.9$ millions trainable parameters.

### A.4 Latent diffusion model training

For the training of the latent diffusion models, the batch size is fixed to $128$. The AdamW optimizer is used with a learning rate of $2 \times 10^{-4}$ and a weight decay of $2 \times 10^{-4}$ through $1,500$ epochs. Similarly to the M2C and BD encoders/decoder, a cosine annealing learning rate scheduler is used with a linear warm-up of $100$ epochs. An EMA with a weight of $0.999$ is also used during testing.

### A.5 More details about the U-Net

The U-Net used in the latent diffusion model follows a standard architecture, designed to work on latent representation of input images. It is made of several residual and attention blocks. Each attention block consists of self-attention on the image features followed by cross-attention with the conditioning embeddings. The encoder path is built from three stages, each containing two residual-attention blocks and a down convolution, where the number of output channels doubles after each stage. A bottleneck with one attention block sandwiched between two residual blocks connects to a symmetric decoder, where down convolutions are replaced by up-convolutions. The U-Net we considered in this work starts with $80$ filters. The attention blocks use height attention heads, and group normalization is used all along the network, with a number of $16$ groups. SiLU functions are used as non-linear activation functions.

At the end, the U-Net counts $\sim 64.4$ millions parameters.

## B More qualitative examples

This appendix presents the qualitative results obtained for the large dataset, presented in Figure 7.

## C Comparison with a simple GAN

As presented in the core body of the paper, we decided to focus on the presentation of our original pipeline. However, prior to this work, we tested several other more standard techniques that did not provide convincing results compared to what we were able to achieve using our transformer-latent diffusion model. For completeness, we present here some details about the GAN-based architecture that we tested, as well as the BDs that we were able to generate using it.

### C.1 Details about the architecture

The architecture we consider is a simple GAN, made of a generator and a discriminator.

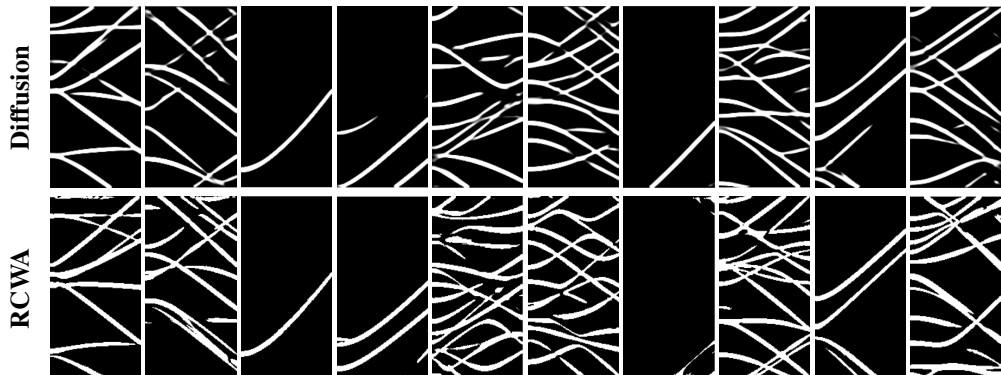

Figure 7: Qualitative comparison of BD generation for the large dataset. The first row shows BDs generated with the diffusion model, and the second one, their corresponding ground truth from RCWA simulations. Samples are randomly selected from the testing set.

On the first side, the generator is made of 9 standard residual blocks, starting with 64 filters in the first block, and doubling the number of filters in each following blocks. Instance normalization is applied after each block along with ReLU activation function. On the second side, the discriminator made of three convolutional layers taking the photonic structure as input as well as the BD produced by the generator. Similarly to what is used for the generator, instance normalization and ReLU activation function is applied after each convolution.

The architectures are trained using the Adam optimizer through a loss composed of a binary cross entropy for the discriminator and an adversarial loss for the generator. The learning rate is fixed to $2 \times 10^{-4}$ and the number of training epochs set to $1,000$. Because we observed a strong tendance to overfitting during our first experiments, we used a dropout of $0.5$. The binary cross entropy is weighted by a factor of $10$ against $1$ for the adversarial loss.

## C.2 QUALITATIVE RESULTS

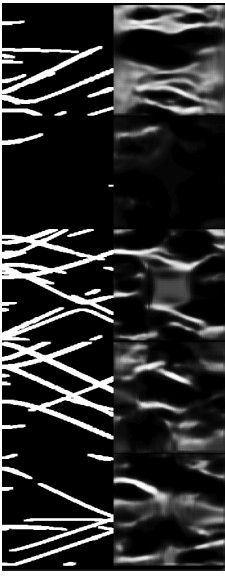

Figure 8: Qualitative comparison of BD generation for the small dataset obtained with a simple GAN pipeline. The first column shows ground-truth BDs generated with RCWA simulations, and the second one, the corresponding BDs obtained with the trained GAN. Samples are randomly selected from the testing set.

