# OpenReview forum: "Towards Photonic Band Diagram Generation with Transformer-Latent Diffusion Models"
_ICLR.cc/2026/Conference — Submitted to ICLR 2026_

### Official Review · Reviewer_uYTB · 2025-10-26

**Soundness:** 3
**Presentation:** 3
**Contribution:** 2
**Rating:** 4
**Confidence:** 3

**Summary:**

The paper introduces a conditional generative pipeline that maps layered photonic structures to their photonic band diagrams. It encodes a stack of two-dimensional slices with a transformer to capture inter-layer interactions, compresses band diagrams into a latent representation via a vision-transformer autoencoder, and synthesizes full diagrams using a latent diffusion model conditioned on the learned structural context. The contributions center on a coherent end-to-end architecture tailored to layered media, a practical data and training setup for this mapping, and internal analyses that illuminate design choices and trade-offs for use as a fast proxy within iterative design workflows.

**Strengths:**

1. **Coherent end-to-end design aligned with the layered-structure setting.** The integration of a slice-wise transformer encoder, a compact latent space for band diagrams, and a conditional generative stage yields a clear information flow; this organization explicitly targets inter-layer coupling while keeping components modular and interpretable, facilitating future substitutions or extensions without altering the overall pipeline.

2. **Practical efficiency for design-loop scenarios.** By generating band diagrams directly from structural inputs, the approach reduces reliance on repeated high-cost simulations and provides rapid feedback during exploration and screening; this predictability and speed are well matched to workflows where timely guidance is more valuable than exact solver-level fidelity at every iteration.

3. **Thoughtful engineering choices and in-scope diagnostics.** The paper includes contrastive pre-alignment between structure and diagram embeddings, ablations on whether to freeze the structural encoder, and qualitative visualizations that reveal success and failure modes; although not a substitute for cross-family baselines, these studies clarify the contribution of each component and improve transparency around training stability and inference behavior.

**Weaknesses:**

1. **Architectural novelty alone does not demonstrate effectiveness.** While the manuscript suggests that prior work relies on dense networks, VAEs, or U-Nets and positions diffusion models as more advanced, this framing does not, by itself, establish superiority; the paper should include controlled, like-for-like comparisons against representative baselines such as a conditional VAE, a deterministic U-Net/Transformer regressor, or other strong generative families trained under matched budgets, with clear reporting of variability and qualitative overlays that reveal where predictions diverge from reference solutions, because only such head-to-head evidence can substantiate the claimed benefits.

2. **The rationale for limited transferability to three-dimensional settings is unconvincing.** The manuscript does not clearly explain why existing approaches could not be adapted to three-dimensional structures, since adding a depth coordinate or layer index is commonly handled via encoding strategies or volumetric operators; consequently, the authors should either provide a conceptual argument for what specifically hinders straightforward 3D extensions in prior methods or, preferably, include comparisons to simple 3D CNN/U-Net/Transformer variants and slice-attention baselines that process the additional dimension directly, as this would clarify whether the proposed pipeline is uniquely capable rather than merely a design preference.

3. **The problem motivation does not isolate a domain-specific challenge that necessitates a tailored method.** As presented, the task appears addressable by a range of generic architectures capable of producing three-dimensional outputs, leaving unclear what special difficulty distinguishes this setting and warrants a bespoke solution; for a venue emphasizing methodological innovation, the paper would benefit from articulating concrete domain hurdles and demonstrating where standard baselines struggle, otherwise the work may be more appropriately positioned in an application-focused forum where its practical utility can be judged against field-specific benchmarks.

**Questions:**

see weaknesses.

---

> ### Author Response · Authors · 2025-11-20
>
> We would like to thank the reviewer for the constructive and detailed feedback of our work, as well as for the positive remarks.
>
> # **Baseline comparisons**
>
> We agree that a broader ablation study would be useful to strengthen the present work and should be considered in a future work. As mentioned in the paper, we explored several alternatives during early development, but in our preliminary experiments these models did not converge to meaningful band diagrams. This absence of convergence in the band diagram is directly related to the difficulty of the task to tackle. The reported architecture was the first that consistently captured both the fine-scale structures of BDs and aspects of the underlying photonic physics. For this reason, it became the focus of the present study. However, to better highlight this, we decided to add some details about a more standard GAN-based architecture that we tested. Those details are presented in Appendix C.
>
> # **Transferability to 3D settings**
>
> We agree that the manuscript could better present the challenges of extending prior approaches to full 3D photonic structures. While adding another dimension is conceptually straightforward, in practice volumetric operators quickly become computationally expensive and struggle to model long-range couplings across deeper stacks. In early experiments, 3D convolutional variants also suffered from either memory limitations or poor convergence, and did not produce convincing band structures, when compared to ground truth. Acknowledging this remark from the reviewer, we added a discussion about those limitations in the paper.
> Our transformer-based representation, by contrast, models the stack as a sequence, allowing long-range optical interactions to be captured through attention regardless of depth. This point is a clear novelty of the present work. That being said, assessing scalability to richer geometries requires generating new datasets, which is computationally intensive and would require several weeks of computation. Within the timeline of the present conference, we are afraid that proper estimations can not be done in time. Therefore, we see this work more as a proof-of-concept, acting as a spark in the community for further discussion and improvement.  We view this remark as an important and required direction for future work and explain that in the manuscript.
>
> # **Domain-specific motivations**
>
> We appreciate the reviewer’s point regarding the need to clarify the specific challenges of this domain, and we consequently add a detailed discussion on that topic in the paper, at the end of Section 3 and in Section 4. Photonic band diagrams present several difficulties that generic architectures struggle with: small pixel shifts lead to large metric deviations ; small changes in one layer can affect the entire diagram globally ; mapping is a highly non-linear process and depends on long-range dependencies that simple convolutions can hardly capture.
>
> These properties make standard surrogate models less pertinent and hard to make converge to an acceptable solution. The proposed approach was the first that consistently handled these complexities and produced physically plausible diagrams. Our contribution is therefore not only architectural novelty, but also demonstrating that “this cross-disciplinary approach is original and timely, bridging modern deep learning architectures with complex photonic design problems.” as acknowledged by reviewer zhRf.
>
> # **Positioning of the model**
>
> We agree with the reviewer that clarifying the intended role of the model is important and would clarify the goal of our work. The proposed surrogate is designed primarily as a fast exploratory tool for iterative design loops, where rapid feedback is crucial and exact solver-level fidelity is not required at every iteration. It is not presented as a replacement for high-precision solvers. Instead, it helps reduce the number of costly RCWA or other simulations by providing meaningful structural insight during exploration. We clarify this point in the revised manuscript, by adding a paragraph in the introduction and strengthening the conclusion into that direction.
>
> ---
>
> We would like to thank the reviewer again for the various feedback. While the current work does not include all suggested baseline comparisons or 3D extensions as required - mainly because of space and time constraints, the present work demonstrates that latent diffusion conditioned on transformer-derived representations is a promising surrogate strategy for modelling the difficult problem of the calculation of photonic band diagrams. We hope this proof-of-concept serves as a solid foundation for broader investigations outlined by the reviewer but will spark discussions and further work in our community.
>
> We also want to mention that we added a diff file with the supplementary materials, for convenience.

---

> > ### Comment · Reviewer_uYTB · 2025-11-25
> >
> > Thank you for the authors’ response. I do not think the rebuttal has addressed the main concerns I raised. It is a well-known fact that GANs are difficult to train, and I understand that the authors may not be GAN experts, so suboptimal training is not surprising. However, VAE is, in principle, more closely related to diffusion models, and it is also faster, cheaper, and easier to train as a baseline. Therefore, not including a comparison with VAE is, in my view, not very reasonable. Overall, I believe the paper should include more well-designed ablation studies to clearly demonstrate the advantages of the proposed method. For these reasons, I have decided not to change my score.

---

### Official Review · Reviewer_zhRf · 2025-11-01

**Soundness:** 3
**Presentation:** 3
**Contribution:** 3
**Rating:** 6
**Confidence:** 4

**Summary:**

This paper introduces an ambitious and conceptually interesting framework for generating photonic band diagrams (BDs) from 3D photonic crystal (PhC) structures using transformer-based latent diffusion models. The approach represents a clear methodological advance over prior deep learning surrogates that relied on CNNs or VAEs, as it leverages transformers to capture long-range electromagnetic couplings and diffusion models to synthesize fine-grained spectral details. The proposed Material-to-Context (M2C) encoder, ViT-based BD autoencoder, and conditional latent diffusion model together form a scalable architecture with the potential to drastically reduce the computational cost of BD prediction—reportedly achieving up to 900× speedup compared to rigorous coupled-wave analysis (RCWA) simulations.

However, the experimental evidence is less convincing than the conceptual innovation. The reported quantitative metrics (Dice ≈ 0.37 on the small dataset and ≈ 0.23 on the large one) indicate limited fidelity, and the generated BDs, while visually plausible, often miss important spectral features. In photonic design, such small spectral mismatches can correspond to large physical deviations, so it remains unclear whether the model’s outputs are reliable enough for practical use. The paper acknowledges these limitations but does not offer concrete solutions, such as uncertainty quantification, physics-informed regularization, or calibration, to mitigate or measure these discrepancies.

**Strengths:**

The paper’s main strength lies in its novel application of latent diffusion models to the problem of photonic band diagram (BD) generation, a domain where traditional solvers like RCWA are computationally intensive. The authors creatively combine transformer encoders and diffusion-based generative modeling, demonstrating how recent advances in generative AI can be repurposed for physics-driven simulation tasks. This cross-disciplinary approach is original and timely, bridging modern deep learning architectures with complex photonic design problems.

Conceptually, using a diffusion process to synthesize physically meaningful spectra represents a significant methodological innovation, offering the potential to replace thousands of numerical Maxwell solves with a single generative inference step. The proposed pipeline, especially the Material-to-Context (M2C) transformer, is thoughtfully designed and technically sound, highlighting an insightful understanding of how self-attention mechanisms can capture multi-layer optical coupling effects.

**Weaknesses:**

1. While the conceptual innovation is clear, the paper’s experimental and analytical depth falls short of its stated ambition. The most significant limitation is the lack of comparison against strong baselines. The authors mention prior CNN-, VAE-, and U-Net–based models for photonic band or dispersion prediction but do not provide direct quantitative comparisons to these architectures. Without such baselines, it is difficult to assess whether the proposed diffusion-transformer framework offers a tangible performance advantage, or whether the observed results could be matched by simpler models.

2. The paper also provides limited discussion on generalization. Although the authors claim that the framework can be scaled to arbitrary 3D photonic structures, all experiments are confined to synthetic, highly regular datasets built from stacked holey and uniform layers. There is no evidence that the model can handle more complex geometries, continuous permittivity variations, or realistic fabrication noise—conditions essential for true generalizability.

3. Another weakness lies in the lack of critical analysis of the model’s underperformance. Reported metrics (Dice ≈ 0.37 on the small dataset, ≈ 0.23 on the large one) and visual comparison are substantially lower than what would constitute reliable physical prediction, yet the paper does not explore the causes. For instance, it remains unclear whether errors arise from the diffusion process, the latent space compression, or the transformer conditioning. A deeper ablation study or error decomposition would have clarified the model’s limitations and helped guide future improvements.

4. Finally, while the authors briefly mention possible remedies—such as larger encoders, physics-informed priors, or ensemble sampling—these ideas are presented only as speculation rather than experimentally supported strategies. Overall, the work would benefit greatly from stronger empirical validation, explicit baseline comparison, and systematic error analysis, which would make its claims of physical relevance more convincing and actionable.

**Questions:**

Could you provide quantitative comparisons against standard baselines such as CNNs, VAEs, or U-Nets used in previous photonic band prediction works? This would clarify whether the diffusion-based approach offers a real advantage beyond architectural novelty.

How well does the model generalize to more complex 3D photonic structures beyond the synthetic stacked-layer dataset? Have you tested any out-of-distribution geometries or permittivity ranges to support your scalability claim?

The reported Dice and mAP scores are relatively low. Could you analyze where the errors come from—for example, from the transformer encoding, latent diffusion stage, or reconstruction process?

Given that small spectral deviations can have large physical impacts, have you considered adding uncertainty estimation or confidence measures to assess the reliability of generated band diagrams?

You mention physics-informed strategies as future work. Could you outline more concretely how such priors or constraints might be incorporated into your diffusion pipeline?

Finally, do you view this model primarily as a fast surrogate for exploration or as a physically accurate predictor for design optimization? Clarifying this distinction would help position the contribution more precisely.

---

> ### Author Response · Authors · 2025-11-20
>
> We would like to thank the reviewer for the detailed feedback and the very positive comments in their introduction.
>
> # **Baseline comparisons**
>
> We agree that a broader ablation study would be useful to strengthen the present work and should be considered in a future work. As mentioned in the paper, we explored several alternatives during early development, but in our preliminary experiments these models did not converge to meaningful band diagrams. This absence of convergence in the band diagram is directly related to the difficulty of the task to tackle. The reported architecture was the first that consistently captured both the fine-scale structures of BDs and aspects of the underlying photonic physics. For this reason, it became the focus of the present study. However, to better highlight this, we decided to add some details about a more standard GAN-based architecture that we tested. Those details are presented in Appendix C.
>
> # **Generalization to more complex geometries**
>
> Our objective here is more to test whether transformer conditioning and diffusion can model BDs in a controlled and restricted setting, hence having a proof-of-concept. However, the pipeline itself is general, requiring only a sequence-of-slice as input, just like the simulation method.
> Generalization to richer structures will require new datasets. Generating large and diverse RCWA datasets is computationally demanding and doing so would represent several weeks of calculations which is beyond the scope of the present proof-of-concept paper. We believe that the cross-disciplinary approach which was acknowledged by the reviewer as “original and timely, bridging modern deep learning architectures with complex photonic design problems” will spark debate and further work in the community.
>
> # **Low Dice/mAP scores and error sources**
>
> We fully agree that the reported dice and mAP values are weak if in the context of a standard computer vision task. Nevertheless, this apparent drawback is relatively expected for this specific problem given the intrinsic structure of photonic BDs: bands are very thin, sometimes only a few pixels wide, so even slight spatial shifts cause the metrics to drop significantly (Joannopoulos et al., 2011). As discussed in the paper, we use these metrics primarily for relative model comparison.
>
> # **Uncertainty**
>
> We agree that uncertainty estimation is important for assessing the physical reliability of generated BDs. Diffusion models naturally support uncertainty estimation through sampling variability: by generating multiple band diagrams from different random initializations, one can construct a confidence or variance map that highlights regions where the model is more or less certain. Taking that remark into account, we added details about this perspective in Section 6 of our revised version of the manuscript.
>
> # **Physics-informed**
>
> As noted by the reviewer, we mentioned adding physics-informed priors, as well as the related literature references, as a future direction. We strongly believe that the modularity of our pipeline makes it well suited for integrating such priors but we leave it to the community/further work by lack of space and time in the present study.
>
> # **Positioning of the model**
>
> We appreciate the reviewer’s request for clarification. The present model is intended primarily as a fast surrogate for exploratory design, enabling rapid exploration of structures before needing full, rigorous and resources costly simulations. It is clearly not positioned as a direct replacement for high-precision solvers, especially in contexts where small spectral deviations have large physical implications. Instead, it provides a computationally efficient first approximation that reduces the overall exploration cost. Based on that comment, we add more details about this point in the revised version of our manuscript.
>
> ---
>
> In summary, we would like to thank the reviewer for the constructive feedback. While this paper does not address all suggested empirical analyses or generalization studies, mainly by lack of time and space, we believe the paper demonstrates that diffusion models conditioned on transformer embeddings are a promising tool for photonic BD surrogate modelling and, as stressed by the reviewer, ”using a diffusion process to synthesize physically meaningful spectra represents a significant methodological innovation, offering the potential to replace thousands of numerical Maxwell solves with a single generative inference step”. Such novel application of latent diffusion models to the problem of photonic BD generation, using a “thoughtfully designed and technically sound” proposed pipeline including a Material-to-Context transformer understanding of how self-attention mechanisms can capture multi-layer optical coupling effects, will, in our humble opinion, spark debate and interest in the community.
>
> We also want to mention that we added a diff file with the supplementary materials, for convenience.

---

### Official Review · Reviewer_UX6z · 2025-11-02

**Soundness:** 3
**Presentation:** 3
**Contribution:** 3
**Rating:** 4
**Confidence:** 4

**Summary:**

The paper proposes a surrogate model for photonic band diagram (BD) generation using a Transformer–Latent Diffusion framework. The method takes a stack of dielectric layers as input, encodes it with a transformer-based material-to-context encoder (M2C), and conditions a latent diffusion model (U-Net backbone) to generate corresponding BDs. The encoder–decoder pair is contrastively trained (MoCo) before fine-tuning the diffusion generator. Experiments use synthetic RCWA-generated datasets of 2–8 layer photonic crystals, showing up to 82× faster inference than RCWA.

**Strengths:**

Strengths:
* Applying diffusion and transformer conditioning to photonic simulation seems to be new.
* The paper is clear and easy to follow

**Weaknesses:**

Weaknesses:
* Poor justification for transformer + diffusion choice. The core question remains unanswered: why should this task need transformers or latent diffusion? The paper never compares against simpler or more conventional surrogates, showing only the adopted model arch's accurcay. Showing the reasons why we need diffusion would give readers more insight about the paper?
* Quantitative quality is very low. Table 2 (p. 8) reports mAP only ≈ 0.23 – 0.44 and dice ≈ 0.23 – 0.37 — poor even for the small dataset, with further degradation on the larger one.
* Limited dataset, especially given the precision seems to be bad. I feel the included dataset is already a simplified one, makeing me concern about the performance on more complicated design.

**Questions:**

Minor:
* The authors fails to include those operator-learning approaches for AI4photonics, e.g.,
PIC2O-Sim: A Physics-Inspired Causality-Aware Dynamic Convolutional Neural Operator for Ultra-Fast Photonic Device FDTD Simulation PACE: Pacing Operator Learning to Accurate Optical Field Simulation for Complicated Photonic Devices
* Dataset realism. Can you try a harder design to check your method's accuracy?

---

> ### Author Response · Authors · 2025-11-20
>
> We would like to thank the reviewer for the constructive comments and the positive feedback on the clarity and novelty of the approach.
>
> # **Justification for transformers and diffusion**
>
> We agree with the referee on that point and we actually explored several simpler alternatives (3D CNNs, 3D VAEs, GANs, encoder–decoder models), but in our experiments these models did not converge to meaningful BDs. BDs are sparse, thin, and highly sensitive to multilayer interference effects, hence a difficult problem to tackle. Transformers are better at modeling long-range couplings through attention, and latent diffusion is effective for generating fine and high-frequency structures. Therefore, we believe our novel approach is promising to tackle such difficult problems in photonic. Hencer, to clarify that point, we strengthen the motivations on why using transformers and diffusion models in the revised version of the paper: mainly at the end of Section 3, and in Section 4.
>
> The reported architecture was the first that consistently captured both the fine-scale structures of BDs and aspects of the underlying photonic physics. For this reason, it became the focus of the present study. However, to better highlight this, we decided to add some details about another more standard GAN-based architecture that we tested previously, as well as an illustration of the band diagrams that we were able to achieve using it. Those details are referenced in the main body of our paper, and are mainly presented in Appendix C.
>
> While we understand that the above discussion is mainly qualitative, we would like to stress that this work should be seen as a proof-of-concept demonstrating that transformer-based conditioning and latent diffusion can help tackle the difficult problem of physically valid BDs. A more exhaustive architectural comparison is indeed needed in the next step, but we think it is beyond the scope and time frame of this initial study. We strongly believe that the presented pipeline is original and innovative for the scientific community, which motivated its compact presentation in this paper.
>
> # **Quantitative results**
>
> We agree that the Dice and mAP scores appear weak from a classical computer vision point of view. However, BD features are only a few pixels thick (Joannopoulos et al., 2011), and small shifts reduce these metrics significantly, hence the challenge of the task. We use them here mainly for comparison purposes. Qualitative inspection is at the moment more informative for assessing physical plausibility. This point raised by the reviewer is clarified in the revised version of the paper, in Section 6, when discussing the quantitative results.
>
> # **Dataset complexity**
>
> We agree with the reviewer on the fact that the dataset is simplified. As explained above, the goal of this first study is to determine whether transformer conditioning and diffusion can generate reasonable BDs and shall be seen as a proof-of-concept. Even in this simplified setting, the mapping is challenging due to the sensitivity of BDs to various geometric and material changes, stressing once more the complexity of the task to handle.
> The pipeline itself is general enough and can, without loss of generalities, handle more complex geometries. Generating large and diverse RCWA datasets is computationally demanding and was intentionally left out of the scope for this initial study, seeking for sparking the debate and interest for future works in the community.
>
> # **Related operator-learning works**
>
> We thank the reviewer for pointing out PIC2O-Sim and PACE. These works are now cited in the revised version, in the related work section. They focus on optical field simulation rather than BD generation but are nevertheless fully relevant to AI for photonics.
>
> # **Harder designs**
>
> We thank the reviewer for pointing out this minor comment for future work. Indeed, training on more complex structures is an important aspect of foreseen future work. This first study aims to establish that a diffusion-based surrogate can produce plausible band diagrams and offer significant speedups. Extending the method to richer geometries will require new datasets. However, the estimated time to push this further is estimated to several weeks (for RCWA simulations and the training of our pipeline) which  is out of reach within the present timeline.
>
> ---
>
> All in all, we appreciate the reviewer’s constructive feedback. While this paper does not explore all the suggested comparisons or harder datasets, mainly due to time constraints, we believe this work already provides a solid foundation for the broader investigations suggested and that, as pointed out by the reviewer, those new results of applying diffusion and transformer conditioning to photonic simulation, in a clear and easy to follow paper, will spark discussion and future work within the community. We also want to mention that we added a diff file with the supplementary materials, for convenience.

---

### Official Review · Reviewer_YhcL · 2025-11-02

**Soundness:** 3
**Presentation:** 3
**Contribution:** 2
**Rating:** 4
**Confidence:** 2

**Summary:**

The paper proposes a learned surrogate to generate photonic band diagrams (BDs) from layered periodic structures far faster than classical solvers. The pipeline assembles: 1. a Transformer encoder over slice sequences of the structure (with cumulative‑depth and Fourier cues), 2. a ViT‑VAE that embeds BD images, 3. contrastive alignment between the two embeddings, and 4. a latent diffusion model conditioned via cross‑attention on the structure embedding to produce the BD. Datasets are synthetic layered stacks; the surrogate achieves large speedups and moderate image‑overlap fidelity. The intended use is rapid screening in design loops.

**Strengths:**

- Clear end‑to‑end generative surrogate with a plausible conditioning pathway (contrastive‑aligned structure embedding feeding a latent diffusion sampler).
- Practical representation of layered structures as sequences with simple positional and spectral cues.
- Demonstrated computational savings that can enable rapid screening workflows.

**Weaknesses:**

- Limited ablations make it difficult to isolate which architectural choices matter (e.g., need for contrastive pretraining vs. direct conditioning; choice of ViT‑VAE vs. CNN; freezing vs. joint fine‑tuning of the structure encoder; where and how conditioning is injected).
- Evaluation leans on image overlap metrics; task‑target mismatch makes it harder to connect score improvements to practically useful BD characteristics.
- Generalization beyond the training manifold (richer geometries, different distributions) is not demonstrated; robustness and uncertainty are not quantified.
- Reporting is light on parameter counts, compute, training stability, and quality‑vs‑cost trade‑offs (number of diffusion steps, guidance scale, etc.).

**Questions:**

1. **Contrastive alignment:** What is the temperature, queue size, and negative sampling strategy? How much does contrastive pretraining contribute versus training the diffusion model end‑to‑end without it? Please provide a clean ablation.
2. **Structure encoder design:** How sensitive are results to depth/width, attention heads, positional encoding (learned vs. sinusoidal), cumulative‑depth encoding, and Fourier channels? A table isolating each component would clarify necessity.
3. **Conditioning pathway:** Where is conditioning injected into the latent diffusion U‑Net (which blocks, which resolutions)? Have you compared cross‑attention with FiLM/adapters/concatenation, and classifier‑free guidance versus conditional‑only training?
4. **BD representation:** Why a ViT‑VAE for BD embedding rather than a CNN or a 1D sequence model along k with local convs across frequency? Please report reconstruction quality, KL weight, latent dimensionality, and how these affect downstream generation.
5. **Freezing vs. joint training:** You note freezing the structure encoder can help. Can you show curves or a table comparing joint fine‑tuning, partial unfreezing, and full freezing, and discuss stability/overfitting issues?
6. **Baselines and simplicity:** How does a direct conditional UNet (no VAE, no contrastive pretraining) compare under the same compute? Similarly, how does a simpler CNN encoder for structures fare against the Transformer?
7. **Robustness and uncertainty:** Can you report sample‑to‑sample variance (multiple diffusion draws) and whether it correlates with errors? Even simple ensembles or variance maps would help quantify uncertainty.
8. **Data splits and leakage:** How are splits defined to avoid near‑duplicate structures across train/val/test? Any augmentations that risk leakage (e.g., deterministic ordering) should be clarified.

---

> ### Author Response · Authors · 2025-11-20
>
> We would like to thank the reviewer for the constructive feedback. We appreciate both the positive remarks and the detailed suggestions for improving the empirical analysis.
>
> # **Limited ablations**
>
> We agree that a broader ablation study would be useful to strengthen the present work and should be considered in a future work. As mentioned in the paper, we explored several alternatives during early development, but these models did not converge to meaningful band diagrams. This is directly related to the difficulty of the task to tackle. The reported architecture was the first that consistently captured both the fine-scale structures of BDs and aspects of the underlying photonic physics. For this reason, it became the focus of the present study. However, to better highlight this statement, we decided to add some details about another more standard GAN-based architecture that we tested previously, as well as an illustration of the band diagrams that we were able to produce using it. This extra analysis detailed in Appendix C.
>
> # **Evaluation metrics and task mismatch**
>
> We acknowledge that overlap-based metrics such as Dice and mAP are not ideal for the present task. Since BDs contain thin, high-frequency features, even small spatial shifts significantly reduce these scores. We use them mainly for comparison purposes. This is clarified in the revised version, in Section 6.
>
> # **Generalization**
>
> We agree that broader generalization is not demonstrated here. Our contribution is a general pipeline that can, in principle, accommodate richer geometries, since it only requires a sequence-of-slices representation. Training on substantially more diverse datasets is computationally expensive and was left intentionally out of the scope for this proof-of-concept. Yet, we consider the generalization as an important next step without diminishing the interest of the community into our novel technique and spark the debate.
>
> # **Reporting parameters**
>
> To respect the space limits, detailed hyperparameters and training settings are provided in the Appendix A, including parameter counts, diffusion steps, and architectural details. We expanded Appendix A to improve clarity and reproducibility. We also highlight that the training code is provided as supplementary material, which we believe is an important aspect for reproducibility and further work by the community, based on this seminal work.
>
> # **More specific questions**
>
> **Contrastive alignment:** we agree with the reviewer’s remark, therefore details on temperature, memory queue size, and negative sampling follow MoCo-style momentum contrast are now reported in Appendix A. Empirically, we observed that contrastive performance correlated strongly with final BD quality. We emphasize those additional information in the revised version of the paper.
>
> **Structural encoder design and ViT-VAE:** we agree that an ablation of cumulative-depth encoding, Fourier channels, attention heads, and model size would be valuable to further strengthen the present work. However, contrastive training takes several days per configuration, the total training time being estimated to several weeks, making such a study infeasible within the review cycle. Regarding the ViT-VAE, it was adopted after observing a slightly higher reconstruction fidelity than a purely convolutional VAE. Other encoders such as more standard VAE may also be suitable.
>
> **Conditioning:** as described in Appendix A, structural embeddings are injected through cross-attention blocks, and time-step embeddings through residual paths. We did not test alternative conditioning mechanisms such as FiLM or classifier-free guidance but those are interesting suggestions that we left for future work.
>
> ---
>
> Regarding the remaining questions, we again agree with the reviewer and appreciate those meaningful recommandations. However, each one of them requires to run additional experiments that we can not afford for now. Even though we cannot reasonably address all the relevant points raised by the reviewer within the current paper and timeframe, we fully understand and appreciate the feedback. We believe our work demonstrates that latent diffusion conditioned on structural embeddings is a viable and promising direction for photonic surrogate modelling. The methodology, dataset, and code provide a solid foundation for the more extensive investigations proposed by the reviewer. We hope that this present study, presenting a clear end-to-end generative surrogate with a plausible conditioning pathway and demonstrating computational savings as pointed out by the reviewer, will spark further discussions and research within the community.
>
> We also want to mention that we added a diff file with the supplementary materials, for convenience.

---

### Meta-Review · Area_Chair_4EP3 · 2026-01-06

**Summary:**

This paper works on photonic band diagrams generation. Authors introduced an method for BD generation based on diffusion models. Authors also provide insights into why transformers and diffusion models are well suited to capture complex interference and scattering phenomena inherent to photonics.

This paper got three 4 ratings and one 6 rating.

The strength of this paper given by reviewers are:
1. Clear end‑to‑end generative surrogate with a plausible conditioning pathway. (Reviewer YhcL)
2. Practical representation of layered structures as sequences with simple positional and spectral cues. (Reviewer YhcL)
3. Demonstrated computational savings. (Reviewer YhcL)
4. Applying diffusion and transformer conditioning to photonic simulation seems to be new. (Reviewer UX6z)
5. The paper is clear and easy to follow. (Reviewer UX6z)
6. novel application. (Reviewer zhRf)
7. thoughtfully designed and technically sound. (Reviewer zhRf)
8. Coherent end-to-end design aligned with the layered-structure setting. (Reviewer uYTB)
9. Practical efficiency for design-loop scenarios. (Reviewer uYTB)
10. Thoughtful engineering choices and in-scope diagnostics. (Reviewer uYTB)


The weakness of this paper given by reviewers are:
1. Limited ablations make it difficult to isolate which architectural choices matter. (Reviewer YhcL)
2. Evaluation leans on image overlap metrics; task‑target mismatch makes it harder to connect score improvements to practically useful BD characteristics. (Reviewer YhcL)
3. Generalization beyond the training manifold (richer geometries, different distributions) is not demonstrated; robustness and uncertainty are not quantified. (Reviewer YhcL)
4. Reporting is light on parameter counts, compute, training stability, and quality‑vs‑cost trade‑offs (number of diffusion steps, guidance scale, etc.). (Reviewer YhcL)
5. Poor justification for transformer + diffusion choice. (Reviewer UX6z)
6. Quantitative quality is very low. (Reviewer UX6z)
7. Limited dataset. (Reviewer UX6z)
8. the paper’s experimental and analytical depth falls short of its stated ambition. (Reviewer zhRf)
9. limited discussion on generalization. (Reviewer zhRf)
10. lack of critical analysis of the model’s underperformance. (Reviewer zhRf)
11. Overall, the work would benefit greatly from stronger empirical validation, explicit baseline comparison, and systematic error analysis. (Reviewer zhRf)
12. Architectural novelty alone does not demonstrate effectiveness. (Reviewer uYTB)
13. The rationale for limited transferability to three-dimensional settings is unconvincing. (Reviewer uYTB)
14. The problem motivation does not isolate a domain-specific challenge that necessitates a tailored method. (Reviewer uYTB)

questions:
1. Contrastive alignment. (Reviewer YhcL)
2. Structure encoder design. (Reviewer YhcL)
3. Conditioning pathway. (Reviewer YhcL)
4. BD representation. (Reviewer YhcL)
5. Freezing vs. joint training. (Reviewer YhcL)
6. Baselines and simplicity. (Reviewer YhcL)
7. Robustness and uncertainty. (Reviewer YhcL)
8. Data splits and leakage. (Reviewer YhcL)
9. fails to include those operator-learning approaches for AI4photonics. (Reviewer UX6z)
10. Dataset realism. (Reviewer UX6z)
11. adding uncertainty estimation or confidence measures? (Reviewer zhRf)
12. outline more concretely how such priors or constraints ? (Reviewer zhRf)
13. model primarily as a fast surrogate for exploration or as a physically accurate predictor for design optimization? (Reviewer zhRf)

In the discuss, Reviewer uYTB replied and mentioned authors didn't address their concern and decided to keep their rating 4.

AC carefully read authors' paper, reviewers' comments and authors' rebuttal. AC found authors postpone a lot weakness to future work and didn't address reviewers' concern (details in the session below). Given this it is hard for AC to make a positive decision and has to reject this paper.

**Reviewer Concerns:**

weakness 1. authors added some work on GAN-based architecture in Appendix C and decided to put other ablation studies in future work.

weakness 2. authors agreed that Dice and mAP is limited.

weakness 3. authors agreed that generalization is not demonstrated by them and decided to leave it for future work.

weakness 4. authors mentioned detailed parameters is reported in appendix A.

weakness 5. authors mentioned other simpler alternative doesn't work. but didn't provide details except standard GAN-based architecture included in Appendix C.

weakness 6. authors agreed Dice and mAP scores is low. but explained BD features are only a few pixels thick made the things hard. But authors should consider some other metrics?

weakness 7. authors agreed dataset is simple, and consider more complicated dataset as future work.

weakness 8. authors mentioned other simpler alternative doesn't work. but didn't provide details except standard GAN-based architecture included in Appendix C.

weakness 9. authors mentioned this work beyond the current paper's scope.

weakness 10. authors agreed Dice and mAP scores is low. but explained BD features are only a few pixels thick made the things hard. But authors should consider some other metrics?

weakness 11. authors didn't provide answer to this.

weakness 12. authors mentioned other simpler alternative doesn't work. but didn't provide details except standard GAN-based architecture included in Appendix C.

weakness 13. authors mentioned they could add more experiments due to limited time and will put this as future work.

weakness 14. authors add a detailed discussion on the specific challenges of the domain.

question 1. authors mentioned they provided details in appendix A.

question 2. authors mentioned it is not infeasible to conduct experiments during review cycle.

question 3. authors mentioned they provided some details in Appendix A. for alternative conditioning, they put it as future work.

question 4-8. authors mentioned they will consider them as future work.

question 9. authors added related work in revised version.

question 10. authors decided to put this as future work.

question 11. authors added this in section 6.

question 12. no concrete steps added.

question 13. authors will add more details in the revised paper.

**Reviewer Scores:**

Reviewer YhcL will keep their score 4.

Reviewer UX6z will keep their score 4.

Reviewer zhRf will keep or lower their score 6.

Reviewer uYTB mentioned they will keep their score 4.

---

### Decision · Program_Chairs · 2026-01-26

Reject